# Suicidal Ideation in Iraqi Medical Students Based on Research Using PHQ-9 and SSI-C

**DOI:** 10.3390/ijerph20031795

**Published:** 2023-01-18

**Authors:** Ahmed Al-Imam, Marek A. Motyka, Beata Hoffmann, Safwa Basil, Nesif Al-Hemiary

**Affiliations:** 1Department of Computer Science and Statistics, Doctoral School, Poznan University of Medical Sciences, Rokietnicka 7 St. (1st Floor), 61-806 Poznan, Poland; 2Department of Anatomy and Cellular Biology, College of Medicine, University of Baghdad, Baghdad 10047, Iraq; 3Barts and the London School of Medicine and Dentistry, Queen Mary University of London, London E1 2AD, UK; 4Institute of Sociological Sciences, University of Rzeszow, 35-959 Rzeszów, Poland; 5Institute of Applied Social Sciences, University of Warsaw, 00-927 Warsaw, Poland; 6Department of Psychiatry, Baghdad Medical City, Baghdad 10047, Iraq; 7Department of Psychiatry, College of Medicine, University of Baghdad, Baghdad 10047, Iraq; 8Psychiatry Council, The Iraqi Board for Medical Specializations, Baghdad 10047, Iraq

**Keywords:** neurotic disorders, patient health questionnaire-9, prevalence studies, scale for suicide ideation, structural equation modeling

## Abstract

Suicidal ideation is a spectrum of contemplations, wishes, and preoccupations with suicide. Its prevalence is ambiguous in Iraq, especially among the youth. We aim to survey the prevalence of suicidal ideation among Iraqi students and explore its risk factors. We surveyed Iraqi undergraduate medical students (*n* = 496) using two psychometric tools, the PHQ-9 and Beck’s SSI-C. We also explored potential risk factors, including the students’ attributes, socio-demographics, and history of mental illnesses. The current study included males (23.8%) and females (76.2%) in their early twenties (21.73 ± 0.11). Concerning PHQ-9 and SSI-C, most students had either moderate (28%) or mild depression (27.8%), while those with suicidal ideation accounted for an alarming 64.9%. The strongest association existed between the SSI-C and PHQ-9 scores (*p* = 0.001, OR = 4.70). Other associations existed with the personal history of mental illness (*p* < 0.001, OR = 2.87) and the family history of suicidality (*p* = 0.006, OR = 2.28). Path analysis highlighted four suicidal ideation predictors, including the PHQ-9 score (standardized estimate = 0.41, *p* < 0.001), personal history of mental illness (0.16, *p* < 0.001), previous psychiatric consultation (0.12, *p* = 0.002), and family history of suicidality (0.11, *p* = 0.005). Suicidal ideation is highly prevalent among Iraqi students. Univariable testing, multivariable analyses, and structural modeling yielded congruent results. The students’ inherent rather than inherited attributes influenced the phenomenon the most, which is in harmony with Durkheim’s theory on the social roots of suicide. We encourage psychiatrists and psychology counselors to be vigilant concerning these risk factors among potential suicidal ideation victims.

## 1. Introduction

Suicidal ideations, also known as suicidal thoughts or ideas, represent a spectrum of thoughts, contemplations, and preoccupations with killing oneself and may progress to attempted or completed suicide [1]. Suicide is among the leading preventable causes of death among the youth, the elderly, and other vulnerable populations [2]. Suicidal ideation involves a hierarchy of feelings from the thinking that “*Life is not worth living*” to more drastic thoughts and contemplated suicide planning, which is a critical issue because most victims of attempted suicides (parasuicides) and completed suicides have acted upon pre-existing ideation [3]. According to Dugas et al. (2012), one-third (34%) of people with lifetime suicide ideation make a suicide plan, and 72% of those with a plan make a suicide attempt, while 26% of those without a plan make an unplanned attempt; most suicide attempts occur within the first year from the onset of ideations [4].

The annual incidence of suicide worldwide is 11.4 per 100,000. However, data records can wrongfully categorize suicides as accidents, homicides, or unknown causes of death [5]. Globally, suicide is a notable cause of death, and it accounts for around 30,000 deaths annually in the United States and more than 5000 in South Africa [6]. According to Lim et al.’s (2019) meta-analysis, the aggregate lifetime and 12-month prevalence of suicidal ideation were 18% and 14.2%, respectively [7]. Parasuicide and suicidal ideation represent taboos for most cultures. Identifying, preventing, and managing victims can be challenging, especially among the young population, which represents the stratum that suffers the most due to the societal stigma and because they may not find a haven to converse freely about the subject. Therefore, healthcare professionals should expand their knowledge of etiopathological bases, including the underlying cognitive mechanisms and socio-demographic risk factors leading to suicidal ideation [8].

One of the oldest definitions of suicide, which dates back to 1897, was created by French philosopher and sociologist Émile Durkheim, while other scholars defined suicide as “*any death that is the direct or indirect result of an act or omission committed by a victim who is aware of the consequences of his action*” [9]. This definition refers to the fatal effect of suicide while distinguishing it from a suicide attempt, which does not always end in death. According to Durkheim (1897), suicide results primarily from social factors, and four types of suicide exist: (a) egoistic (poor integration into society); (b) altruistic (over-integration into society, such as in political hunger strike); (c) anomic (loosening bonds between people, for instance, in the inner city); (d) fatalistic (excessive regulation by society and no personal freedom, for example, the suicide of enslaved people) [9]. Edwin Shneidman’s definition (mid-20th century) takes into account cultural–geographical factors, and it defines suicide as a behavior that is a consciously undertaken action aimed at self-destruction; the action embodies a multidimensional disorder occurring in an individual whose needs are not satisfied and who defines himself as the problem, in which he perceives suicide as the best solution [10]. Each of the former two definitions implies an awareness of suicidal acts and their consequences.

Brunon Hołyst presented in his definition the complexity of the problem of suicide in terms of motives, processes, and consequences. According to the Polish suicidologist, “*suicide was a cultural form of solving life problems according to external dictates, […] a social form of exclusion from the circulation of benefits […], a form of psychological disapproval of a certain form of life […], a biological form of escape from pain, an ideological form of rebellion against the inevitability of death, which can be made an act of choosing the time, place and manner of departure from the world of the living.*” [11]. These former aspects point out that the act of suicide is the final link in the process of self-destruction that sometimes lasts for years, in which suicide is not treated here as a process but an event: “*it is not just a case of tragic self-destruction, but a sequence of interconnected thoughts and actions sometimes lasting for years*” [12].

Clinical psychologist and suicidologist Edwin Shneidman authored a unique typology of suicide attempts, the fundamental core of which is the “*risk of death*” concept. Shneidman defines it as the probability of an individual committing suicide at a given time or soon in the future [13]. On this basis, he distinguished the following types of suicide: (1) simulated suicides—without death, with a shallow risk of death, e.g., posing as a suicide death, disappearing from the place of residence, which can trigger suicide; (2) pseudo-suicides—with a low risk of death, or without death, e.g., taking a harmless substance to a fake poison; (3) parasuicides—with a moderate risk of death, e.g., self-harm, aimed not at death itself, but at changing life situations, or an improving relationship with a significant person; (4) ambiguous suicides—with moderate to high risk of death, e.g., some accidents and homicides provoked by the victim. This group includes people with an ambivalent attitude toward life and death. (5) Suicide attempts—with a high risk of death. These are behaviors aimed at death but ending with the accidental rescue of the individual; (6) suicides—behaviors with a high risk of death, ending in death.

Nonetheless, the former classifications do not exhaust the rich typological issues. It is possible to distinguish typologies due to the will and degree of certainty of death, motivation, attitude to death, awareness of the choice of death, or the degree of interpersonal relationships present in society. Risk factors may also relate to these typologies, which can vary from one population to another based on culture and geography, and these factors relate to the patient’s demographics, psychological and physical illness-related factors, and mental state factors. Approximately 90% of people who commit suicide suffer from a psychiatric disorder, and approximately 50% of those suffer from depression, 25% from alcoholism, 5% from schizophrenia, and 20% from other disorders (personality disorders, chronic neurotic disorder, psychoactive substance abuse, among others) [14].

The current study uses reliable psychometric tools to survey the prevalence of suicidal ideation. The implementation of validated questionnaires is crucial for reliable and replicable studies. We also emphasize the importance of early detection of suicidal ideations and parasuicides, as these represent critical scenarios that touch the essence of human existence. Healthcare professionals cannot make early detection without using validated psychometric tools and robust inferential analyses to identify significant risk factors and predictors of suicidal ideations, parasuicides, and suicides. To the best of the authors’ knowledge, our study is the first from Iraq, aiming to explore risk factors and potential predictors of suicidal ideation among Iraqi undergraduate medical students who live in Baghdad (the Iraqi Capital) and other governorates.

## 2. Materials and Methods

The ethics committee of the psychiatry council of the Iraqi Board for Medical Specializations approved the study (protocol on the 22 February 2022). The current study abided by the Declaration of Helsinki of the World Medical Association and the ethical principles of the Framingham consensus of 1997. We obtained informed consent from each study participant.

The current study is cross-sectional, and the researchers conducted it among undergraduate medicine, dentistry, and pharmacy students from Iraq. The statistician calculated the sample size for a population proportion using the finite population formula while considering the prevalence of suicidal ideation (14.2%) that Lim and coworkers reported in their meta-analysis, the confidence interval for hypothesis testing (95%), and the margin of error (5%) [7]. The sample size calculation mandated a minimum of 385 individuals. Nevertheless, we accessed a larger sample and recruited 496 students.

We implemented two psychometric tools (instruments): the Patient Health Questionnaire-9 (PHQ-9) and Beck’s Suicide Ideation Scale-Current (SSI-C) [15,16]. Researchers use the PHQ-9 to diagnose and categorize the severity of depression; its score ranges from 0 to 27, and it categorizes depression into five categories: minimal depression (1–4), mild (5–9), moderate (10–14), moderately severe (15–19), and severe depression (20–27) [15]. The SSI-C instrument measures the current suicide ideation; its score ranges from 0 to 38, and the higher the score, the stronger the transition to higher-risk suicide ideation [16].

According to de Beurs and coworkers (2015), research indicated that the best cut-off to indicate high/low-risk suicide ideation was when the score was more than two [17]. However, Beck’s cut-off point relied on ROC analysis for maximum sensitivity and specificity. Beck and collaborators declared: “*Raw scores of 2 or higher on the SSI-C and scores of 16 or higher on the SSI-W were described as higher risk*”. Beck’s SSI-C had an odds ratio of 5.42 (95% CI: 2.63–11.17), a sensitivity rate of 53%, and a specificity rate of 83%. On the other hand, it appeared that de Beurs cited another study (secondary referencing), which adopted a slightly higher cut-off (by one point) than Beck’s [18]. Nonetheless, we used Beck’s cut-off point for the current study because we opine that it is more reliable per the critical analysis of both studies. In addition, Beck’s study included a substantial number of participants (*n* = 3701) who were followed up over 20 years between 1975 and 1994. Beck and coworkers (1999) indicated that a raw total score of 2 or higher indicates a higher risk (people with suicide ideation) compared to a raw total score of less than 2 (people without suicide ideation) [18].

We collected other parameters, including age, gender, marital status, residence, college affiliations, year of study, religious affiliations, personal history of chronic illness, history of self-medication, recent stress events, history of prior psychiatric consultation, personal history of mental illness, family history of mental illness, and family history of suicidality. The researchers asked the students if they had encountered recent stress events that could have disturbed their daily routine or quality of life while emphasizing its distinctive nature from psychologically traumatic events. The researchers also asked the students about self-medication without indicating (probing) whether these included illicit drugs or medications that are not over-the-counter medicines.

We tabulated the raw data using Microsoft Excel 2016 and conducted the data analysis using IBM-SPSS and IBM-SPSS Amos version 26. The statistician ran descriptive statistics, correlation matrices, parametric and non-parametric tests (in correspondence with Shapiro–Wilk normality testing), the chi-squared test (or Fisher’s exact test), multivariable ordinal regression, and structural modeling. We considered an alpha (α) value of 0.05 as the cut-off margin for statistical significance in hypothesis testing while implementing the Bonferroni correction to adjust the *p*-value calculations in multiple tests.

## 3. Results

### 3.1. Students’ Attributes, College Affiliations, and Place of Residence

The total sample included 496 students (*n* = 496) with males (118, 23.8%) and females (378, 76.2%), and a male-to-female ratio of approximately one to three (1:3.2). Concerning age, we calculated the mean and the standard of the mean (21.73 ± 0.108, 95% CI = 21.52 to 21.94). There was no significant difference between males and females concerning age (21.75 ± 0.296 versus 21.73 ± 0.108, *p* = 0.437). We also compared age across the five categories under the PHQ-9, but there was no significant difference (*p* = 0.093), and there was also no significant difference in age between people with suicide ideation and people without suicide ideation (*p* = 0.109). Almost half of the students were affiliated with medical schools (47.6%), while others studied dentistry (30.6%) and pharmacy (21.8%). Most were single (95.6%), while the rest were married. Most students were Muslims (96.8%), while the remaining were either Christians or practiced other religions. One-third of the students were enrolled in the fifth year of their studies (33.3%), while the remaining students were enrolled in the first year (16.7%), second (16.3%), third (12.3%), fourth (12.1%), and sixth (9.3%). Most inhabited Baghdad (85.7%) and Diyala (5.4%), while the remaining students resided in other Iraqi governorates (see Appendix A). Some students had a previous psychiatric consultation (7.1%), a personal history of mental illness (27%), and a personal history of chronic illness (13.1%). Further, some students had a history of self-medication (11.5%), and many had encountered recent stresses in their lives (91.3%). Concerning the students’ families, some had a family history of mental illness (17.5%) and a family history of suicidality (14.5%). Concerning the PHQ-9, students had either moderate (28%) or mild depression (27.8%), while the remaining students assumed the other three PHQ-9 categories, including moderately severe (19%), severe (16.9%), and minimal depression (8.3%). On the other hand, regarding the SSI-C, almost two-thirds of the students had suicide ideation (64.9%), while the rest were people without suicide ideation (35.1%).

When stratifying the sample according to the subject of studies (college affiliations), more medical students admitted to having previous psychiatric consultation (10.6%) compared to the pharmacy (4.6%) and dentistry students (3.3%). Similarly, more medical students (30.9%) had a history of mental illness compared to dentistry (24.3%) and pharmacy students (22.2%). Further, more medical students (16.5%) had a history of chronic illness compared to the pharmacy (11.2%) and dentistry students (9.2%), while more medical students admitted to having self-medicated (14.8%) compared to dentistry (8.6%) and pharmacy students (8.3%). Students of medicine, dentistry, and pharmacy had a similar burden of recent stress events (90.3%, 91.4%, and 93.5%, respectively). Concerning the family history of mental illness, more pharmacy students (22.2%) had a positive history compared to students of medicine (17.8%) and dentistry (13.8%). Similar results existed concerning the family history of suicidality among the students of pharmacy (18.5%), medicine (14.4%), and dentistry (11.8%). Concerning the PHQ-9, up to two-thirds of students had either moderate or mild depression, but at varying contributions, and almost one-fifth of medicine students were severely depressed (20.3%), which is high compared to students of dentistry (15.8%) and pharmacy (11.1%) (see Appendix A). Concerning the SSI-C, more dentistry students (68.4%) had suicide ideation compared to medicine (64.8%) and pharmacy students (60.2%).

We compared those who lived in Baghdad with the other governorates. More individuals who lived in Baghdad had a prior consultation than those from other governorates (7.3% versus 5.6%). Similarly, more students who lived in Baghdad had a higher frequency of personal history of mental illness compared to other students (28.7% versus 16.9%). However, concerning the history of chronic illness, students from other governorates had a higher proportion than students living in Baghdad (15.5% versus 12.7%). Similarly, those from other governorates had a slightly higher proportion concerning self-medication (12.7% versus 11.3%), but they had a lower percentage of a recent stress event (85.9% versus 92.2%). Concerning the family history of mental illness, students living in Baghdad expressed a higher percentage (18.1% versus 14.1%). Similarly, concerning the family history of suicidality, students from Baghdad were also in the lead (14.8% versus 12.7%). Students from all governorates had mild to moderate depression, but those from Baghdad had a bit higher proportion (56.2% versus 53.5%), while some students were severely depressed, especially those living in Baghdad (18.1% versus 9.9%). The inhabitants of Baghdad had more suicide ideation than the other regions of Iraq (65.6% versus 60.6%).

### 3.2. History of Self-Medication, Recent Stressful Events, and Chronic Illnesses

Concerning the history of self-medication, most students with a positive history were either moderately (28.1%) or mildly depressed (22.8%), while the remaining students had either mild (28.5%) or moderate depression (28%). Students with and without a history of self-medication expressed an almost matching contribution to suicide ideation (66.7% vs. 64.7%). Most students who had not encountered recent stress events had either minimal (37.2%) or mild depression (34.9%). On the contrary, most of those who had encountered recent stresses were either moderately (29.1%) or mildly depressed (27.2%). Further, students who had encountered recent stress events had a higher proportion of suicide ideation than the stress-free group (65.8% versus 55.8%). On the other hand, most students who had a history of chronic illness had either mild (27.7%) or moderate depression (23.1%), while the other students were either moderately (28.8%) or mildly depressed (27.8%). Concerning the SSI-C, students with and without a history of chronic illness had almost similar proportions of suicide ideation (64.6% versus 65%).

### 3.3. Personal History of Mental Illnesses

In students who had a prior psychiatric consultation, we found that most students were either severely (28.6%) or moderately depressed (25.7%), while most of those who had not had previous psychiatric consultations were either mildly (28.4%) or moderately depressed (28.2%). Concerning the SSI-C, students with a prior psychiatric consultation experienced more suicide ideation (71.4% versus 64.4%). Further, most students who had a personal history of mental illness had either severe (35.1%) or moderate depression (23.9%), while the remaining students were either mildly (32.3%) or moderately depressed (29.6%). Concerning the SSI-C, students with a history of mental illness had substantially higher suicide ideation (80.6% versus 59.1%).

### 3.4. Family History of Mental Illnesses and Suicidality

Concerning a family history of mental illness, most individuals with and without a family history had mild to moderate depression (55.2% versus 56%). On the contrary, students with a positive family history experienced more suicide ideation (70.1% versus 63.8%). Concerning the family history of suicidality, most students with a positive history had severe (34.7%) or moderately severe depression (26.4%), while those without a family history were either mildly (30.7%) or moderately depressed (28.8%). Concerning the SSI-C, individuals with a positive family history of suicidality experienced more suicide ideation than other students (79.2% versus 62.5%).

### 3.5. Univariable Hypothesis Testing

We implemented correlation matrices among all study parameters (see Appendix A). We found a significant correlation of moderate effect size between PHQ-9 and SSI-C (correlation coefficient = 0.554, *p* < 0.001). Further, when stratifying the sample based on gender, we found that females had a stronger correlation (0.574, *p* < 0.001) compared to males (0.478, *p* < 0.001). Other significant correlations existed within the non-stratified sample. For example, the PHQ-9 correlated significantly with the history of psychiatric consultation (0.103, *p* = 0.022), personal history of mental illness (0.323, *p* < 0.001), family history of suicidality (0.222, *p* < 0.001), and recent stress events (0.246, *p* < 0.001). The SSI-C correlated significantly with the history of psychiatric consultation (0.101, *p* = 0.025), personal history of mental illness (0.323, *p* < 0.001), family history of mental illness (0.099, *p* = 0.027), family history of suicidality (0.199, *p* < 0.001), recent stress events (0.106, *p* = 0.018), and most importantly with PHQ-9 (0.554, *p* < 0.001). Most of the former correlations possessed mild to moderate effect sizes, and all were in the positive direction of trends. A similar pattern of correlations existed when stratifying the sample based on gender, while those within the females’ cohort indicated a relatively larger effect size.

According to the former statistics, several risk factors might influence the PHQ-9 and SSI-C, especially those related to prior psychiatric consultation, personal history of mental illness, recent stressful events, family history of mental illness, and family history of suicidality. Therefore, we ran the chi-squared test of independence to reach an inference concerning the former assumptions (Table 1). Concerning the previous psychiatric consultation, there was a significant association only with PHQ-9 (χ^2^ = 3.95, *p* = 0.047, OR = 1.99). Regarding the personal history of mental illness, there was a significant association with PHQ-9 (χ^2^ = 39.76, *p* < 0.001, OR = 3.65) and SSI-C (χ^2^ = 19.82, *p* < 0.001, OR = 2.87). However, concerning the history of chronic illness and self-medication, there was no significant association with PHQ-9 or SSI-C. Concerning the recent stress events, there was a significant association strictly with the PHQ-9 (χ^2^ = 12.04, *p* = 0.001, OR = 4.70). Concerning the family history of suicidality, there was a signification association with the PHQ-9 (χ^2^ = 23.29, *p* < 0.001, OR = 3.40) and SSI-C (χ^2^ = 7.507, *p* = 0.006, OR = 2.28). Furthermore, concerning the PHQ-9, there was a significant association with SSI-C (χ^2^ = 48.34, *p* < 0.001, OR = 4.81), which had the largest effect size among all the tested associations.

### 3.6. Analysis of the Ninth Item (Question 9) of the PHQ-9

Concerning the PHQ-9 Questionnaire’s ninth item, the students answered either with “Not at all” (253, 51%), “Several days” (134, 27%), “Nearly every day” (61, 12.3%), or “More than half the days” (48, 9.7%). The distribution of the ninth items, when considering gender, showed lower suicide ideation among males: “Not at all” (56.8% vs. 49.2%), “Several days” (22.9% vs. 28.3%), “Nearly every day” (12.7% vs. 12.2%), and “More than half the days” (7.6% vs. 10.3%). On the other hand, students of medicine expressed higher suicidal ideation compared to students of pharmacy and dentistry concerning “Not at all” (47.3% vs. 54.4%), “Several days” (26.2% vs. 27.8%), “Nearly every day” (14.8% vs. 10%), and “More than half the days” (11.8% vs. 7.7%). Further, the ninth item of the PHQ-9 correlated significantly with the PHQ-9 total score (coefficient = 0.581, *p* < 0.001) and the SSI-C (0.463, *p* < 0.001). The ninth item was significantly associated with the SSI-C per the chi-squared test of independence (χ^2^ = 100.44, *p* < 0.001, OR = 8.44). In summary, the PHQ-9 ninth item indicated that students of medicine and females had a higher frequency of experiencing suicidal ideation, and it correlated significantly with the total score of the patients’ health questionnaire-9 and the measure of current suicide ideation. Students who answered the item with “Yes” had a much higher likelihood of being categorized as people with suicide ideation per Beck’s score.

### 3.7. Multivariable Hypothesis Testing

The former analyses via the chi-square test of independence revealed some significant risk factors that can influence PHQ-9 and SSI-C, including previous psychiatric consultation, personal history of mental illness, recent stressful events, family history of suicidality, and the PHQ-9 score. However, the earlier tests did not account for all variables simultaneously, i.e., they did not represent a multivariable model. Therefore, we ran an ordinal regression incorporating all variables at once. We used two ordinal regression models while considering the PHQ-9 as the outcome (dependent) variable for the first model and SSI-C as the outcome variable for the second model. Concerning the PHQ-9 and SSI-C, we ran two sub-models for each. The first model incorporated all of the variables without transformation, while the second integrated all of the variables following transforming ordinal ones into dichotomous ones to overcome the limited sample size within some subgroups of students.

Concerning the model-fitting information of the non-dichotomized ordinal regression for the PHQ-9, there was an overall significant model (Nagelkerke’s R^2^ = 0.27, *p* < 0.001) that highlighted some significant risk factors, including female gender (*p* = 0.011, OR = 1.65), personal history of mental illness (*p* < 0.001, OR = 3.82), family history of suicidality (*p* < 0.001, OR = 2.67), and recent stress events (*p* < 0.001, OR = 6.96). Concerning the model-fitting information of the dichotomized ordinal regression for the PHQ-9, there was an overall significant model (Nagelkerke’s R^2^ = 0.19, *p* < 0.001). Significant risk factors included the family history of suicidality (*p* < 0.001, OR = 2.89), the personal history of mental illness (*p* < 0.001, OR = 3.22), and recent stressful events (*p* = 0.004, OR = 4.53). Concerning the model-fitting information of the non-dichotomized ordinal regression for the SSI-C, there was an overall significant model (Nagelkerke’s R^2^ = 0.43, *p* < 0.001), and the significant risk factors included the personal history of mental illness (*p* = 0.032, OR = 1.62) and non-Islamic religious affiliation (*p* = 0.049, OR = 2.69). Concerning the model-fitting information of the dichotomized ordinal regression for the SSI-C, there was an overall significant model (Nagelkerke’s R^2^ = 0.19, *p* < 0.001), and the significant risk factors included the personal history of mental illness (*p* = 0.023, OR = 1.95) and the PHQ-9 (*p* < 0.001, OR = 4.22).

### 3.8. Path Analysis and Structural Modeling

Based on the results from the former analyses, we ran a path analysis as part of the structural equation modeling while incorporating four predictors, including previous psychiatric consultations, recent stressful events, personal history of mental illness, and family history of suicidality. We also integrated the PHQ-9 as an intermediary variable between the former four predictors and the SSI-C (Figure 1). Only three predictors had a significant effect on the PHQ-9, including the family history of suicidality (0.19, *p* < 0.001), recent stress events (0.21, *p* < 0.001), and the personal history of mental illness (standardized estimate = 0.28, *p* = 0.001). Concerning the SSI-C, four predictors were significant, including the family history of suicidality (0.11, *p* = 0.005), previous psychiatric consultation (0.12, *p* = 0.002), personal history of mental illness (0.16, *p* < 0.001), and the PHQ-9 (0.41, *p* < 0.001). Our path analysis had good model fitness (χ^2^/df < 3 and RMSEA < 0.05).

## 4. Discussion

In the present study, most students had mild or moderate depression, while almost two-thirds were people with suicidal ideation. The strongest significant association existed between the SSI-C and PHQ-9 scores, while other associations existed with the personal history of mental illness and the family history of suicidality. Path analysis highlighted four suicidal ideation predictors, including the PHQ-9 score, personal history of mental illness, previous psychiatric consultation, and family history of suicidality. Our univariable tests, multivariable analyses, and structural modeling generated harmonious results concerning the risk factors, including the PHQ-9 score, personal history of mental illness, and family history of suicidality. The students’ inherent rather than inherited attributes influenced the suicidal ideation phenomenon the most. We opine that the former results are also in congruence with Durkheim’s theory on the social roots of suicide.

In 2022, suicide is still a significant public health challenge. Globally, the number of people who commit suicide each year is nearly 800,000 [19]. According to data from 2019, there was a one-quarter decline in suicides worldwide between 1990 and 2016; nonetheless, during 27 years of monitoring, the number of deaths by suicide increased by nearly 7%. Suicide was among the top ten leading causes of death in Central and Eastern Europe, the high-income Asia-Pacific region, and the high-income Australasia and North America regions. The highest suicide death rates existed in Lesotho, Lithuania, Russia, and Zimbabwe when considering country trends in 2016. In contrast, researchers found that the lowest rates were in Lebanon, Syria, Palestine, Kuwait, and Jamaica [20].

Admittedly, a link between suicidal behavior and the COVID-19 pandemic has been identified in some countries [21,22,23,24]. However, the factors influencing such radical steps appear to be more complex and represent multiple etiologies and, most importantly, represent a public health problem noted even before the pandemic. Concerning demographic factors, suicide attempts are more common among women than men; however, death attempts are more frequent among men [20,25,26]. Other studies indicate that suicidal thoughts and behaviors are more common among middle-aged and low-income individuals [26]. However, a review by Huang et al. (2017) noted that no specific demographic factors (including gender, marital status, and religious denomination) proved to be significant risk or protective factors. Admittedly, the overall effect of these data as risk factors was statistically significant but possessed a weak effect size. However, the researchers are far from trivializing the role of demographic factors as both risk and protective factors. They recommend that future studies explore less studied demographic factors and advocate for the more accurate reporting of sample characteristics and study outcomes [27].

In 2008, Brent and Melhem, based on a review of numerous publications on suicide, presented evidence of familial transmission of suicidal behavior, particularly emphasizing genetic and environmental aspects. Genetic determinants are associated with an upbringing in dysfunctional families where there was, for example, parental mental illness or violence, sometimes suicide attempts, and premature death of parents. People raised in such environments are more likely to suffer from depressive disorders, use substances to alleviate unpleasant emotional states and, as a result, lower their resistance to stressful situations. In turn, environmental factors are primarily just a history of suicide in the family, and attempts by other members to take their own lives are associated with the imitation of such solutions to stressful situations [28]. A year later, Lizardi et al. (2009) published data from a study conducted on a sample of 190 people after suicide attempts, which found that a positive family history of suicidal behavior is an important indicator to consider when working psychotherapeutically with a depressed patient after previous suicide attempts. The authors found that suicide attempts are significantly more common among those with a family history of suicidal acts than those without such a family history. In addition, it was found that among those with a first-degree family history of suicide, suicide attempts—if unsuccessful—are significantly more likely to recur [29].

The study by Rajalin et al. (2013) also confirmed earlier results obtained in similar studies. In the measure in question, more than 180 people were included after suicide attempts, and the family history of suicidal behavior was examined using the Karolinska Suicide History Interview and by analyzing patient records. The study found that males who had attempted suicide and had a positive family history of suicide made detailed planned suicide attempts and had a significantly higher risk of suicide than those without such experiences. In addition, a significant factor in attempting to take one’s own life was trauma experienced in childhood [30]. Similar results were found in other studies conducted in Denmark [31], France [32], India [33], and Sweden [34], among others. Thus, it should be acknowledged that early environmental risk factors—especially a history of suicidal behavior in the first-degree family—play a crucial role in the etiology of suicidal behavior.

Research findings suggest that more than 90% of people who die by suicide have a psychiatric diagnosis, mainly depression [25]. Unfortunately, personal experience of mental health problems is one of the frequently cited reasons for the occurrence of resignation thoughts as well as attempts to take one’s own life [35,36,37,38]. Considering stressful situations, studies’ most commonly identified stressors that activate suicidal thoughts include intimate partner problems, work problems, financial hardness, physical health problems, family problems, and criminal or legal problems [39]. Despite numerous published studies and research attempts worldwide to determine the causes and methods to help those struggling with complex crisis states, the current situation is not changing significantly. A systematic review of numerous clinical guidelines on suicide has not helped to establish clinical standards for diagnosing, assessing, and treating suicidal thoughts or individuals at risk of suicide [40].

Due to the research sample adopted, which includes medical students, our data can also be compared with data from surveys conducted among medical students in other countries. Miletic et al. (2015) conducted a similar study among Serbian medicine students (*n* = 1296) at the University of Belgrade. Respondents filled out the PHQ-9 questionnaire as in our study. The PHQ-9 scores ranged from 0 to 27, with a mean of 6.21 (±4.96), and the depressive states correlated with the mean scores obtained, age, relationship status, and gender. The study authors reported that the history of suicide attempts correlated with PHQ-9 scores, mean scores, relationship status, gender, history of mental illness, and drug use; further, the authors noted that the prevalence of recognizable indicators of depression among college students is increasing [41].

Another study conducted among medical students at Gondar University in Ethiopia (*n* = 393) found that factors significantly associated with suicidal thoughts included female gender, comorbid depressive symptoms, chewing psychoactive khat, and poor social support. Further, significant factors associated with realizing these thoughts (suicide attempts) included female gender, personal history of depression, and having a family history of mental illness [42]. In a study conducted among Brazilian medical students (*n* = 4840), risk factors included younger medical students, those who suffered from tremendous stress, and even harassment by other students. In addition, the researchers noted that significant risk factors included female gender, childhood trauma, low income, poor quality of friendships and family relationships, depression, substance abuse, and previous suicide attempts [43]. A meta-analysis of risk factors for suicidal thoughts and suicide attempts among medical students found that the most severe risks for suicidal thoughts and suicide attempts were primarily mental health problems, including depression, burnout, co-occurring mental illness, and stress [44].

We also want to emphasize the importance of the results of the current study considering the ethnic and cultural aspects of suicidal behavior. The present study can also be compared with other studies conducted in Muslim countries. Populations from countries practicing Islam are considered collectivist communities [45], which means, among other things, that they can count on the support of relatives and other members of society who provide vital psychological protection in difficult life situations [46]. Novel results were obtained in a study conducted among students from 11 Muslim countries. The study was conducted on an impressive sample of nearly 7500 students, of whom more than one-fifth reported ever having suicidal thoughts, while more than 14% admitted having multiple suicidal thoughts. The rate of suicide attempts among those reporting such thoughts was 34%, while among those who had attempted to take their own lives, not quite half admitted to multiple suicide attempts. However, students who indicated reliable social ties were less likely to report the occurrence of suicidal thoughts than others. In the former study—as in ours—the number of stressful life events strongly predicted suicidal ideation and such behavior [45].

Another study conducted in Iran on a sample of 856 people after suicide attempts found that the occurrence of both suicidal thoughts and previous suicide attempts were predictors of first as well as subsequent suicidal behavior. However, the study’s authors point out other factors that lead to unplanned suicide attempts, such as substance abuse, impulse control disorders, and stressful life situations, among others, and emphasize the role of preventive interventions as a valuable protection against engaging in such behavior. They also point out that due to the cultural stigma surrounding suicide, data on these behaviors can be inaccurate in official reports [47]. It should also be borne in mind that the environmental crises associated with, among other things, the military action that took place in Iraq, especially in the first decade of the 21st century, pose a significant threat to the mental health of the entire society that is engulfed by this crisis [48]. Moreover, in earlier years, under Saddam Hussein, following the economic sanctions imposed on Iraq by the UN in 1990, the quality of life of Iraqis deteriorated. These sanctions contributed to malnutrition, hindrances to the Iraqi health system in many ways, and a lack of safety and health security [49]. The overlap of so many crises has more than likely affected the mental health of Iraqi society.

### 4.1. Protective Factors

Some research initiatives have also sought to establish protective determinants against this behavior. A study in the Netherlands noted that a low risk of suicide existed in people with a high education level, non-Western immigrant backgrounds, and first-generation immigrants who are married, with children, and receive higher incomes that provide economic stability. The study also noted that urban and regional differences were primarily insignificant [26]. A study of Serbian students called for the design of prevention programs targeting depressive symptoms and, above all, providing students with education and preparation surrounding the appropriate mechanisms to deal effectively with pressure during education and education on alternative strategies for coping with stress, especially among younger students, who appear to be at higher risk of depression and suicide [41]. Authors made analogous suggestions for studies among medical students [42,43].

Being religious versus engaging in suicidal behavior that contradicts the professed doctrine is a separate issue. In this situation, the reasons for breaking internalized and hitherto adhered-to rules are particularly relevant to the present study. Islam rectifies a code of conduct for Muslims, indicates values, and sets ethics conducive to the development of adaptive methods of coping with difficult life circumstances and allows them to tolerate these problematic states, teaching them to live in peace. According to Islam, people cannot choose between life and death, and suicide is considered a major sin. Muslims believe that suicidal people are not allowed to enter Paradise. In addition—notably, in countries where Islam is professed—one can see other important protective factors against making such drastic decisions. These include having large families, peers, and ethnic networks, and generally strong ties reducing suicide in multigenerational households [50].

Moreover, in the 2021 study mentioned above, Eskin et al. found that strong social ties in Muslim societies are essential protective factors against suicide attempts despite such ideation [45].

In an analysis of trends in suicidal behavior conducted using data from 46 Muslim-majority countries around the world, it was noted that most of them had lower age-standardized suicide rates than the global average. According to the authors, while these data may suggest that religious beliefs and practices among Muslims influence the failure to engage in such behavior, the data would need to be verified in subsequent studies conducted by independent parties, especially in a research process in which all data would be available to the researchers. In the cited studies, data on deaths were either unavailable or deemed useless due to their poor quality. These results represent WHO estimates based on suicide data obtained from only three countries (Kazakhstan, Kuwait, and Kyrgyzstan) [51].

### 4.2. Study Limitations

The current study has limitations inherent to the cross-sectional design. Another crucial point could be that the current study implemented Beck et al.’s (1999) cut-off point for the SSI-C scale, while de Beurs and coworkers (2015) highlighted another cut-off value [17,18]. However, Beck’s cut-off point relied on ROC analysis for maximum sensitivity and specificity. On the other hand, de Beurs cited Brown et al. (2000), who adopted a slightly higher cut-off than Beck [52]. Subsequent research should include larger samples, implement follow-up strategies among people with suicide ideation via longitudinal studies, incorporate biological sample investigations and psychometric testing, and adapt experimental designs via controlled trials for a superior level of evidence. The heterogeneity of results from the literature mandates robust meta-analyses concerning the prevalence of suicidal ideation and its risk factors among vulnerable populations, including the youth.

It could also be interesting to implement new psychometric tools that consider additional facets of suicidal behavior, such as the SENTIA-Brief scale that Díez-Gómez et al. (2021) examined, and according to the authors is a simple and brief instrument that can be effectively used as a screening tool in clinical and educational settings [53]. Another example is the Paykel Suicide Scale (la Escala Paykel de Suicidio) that Fonseca-Pedrero and Pérez de Albéniz (2020) reported on, and they concluded that it is short and straightforward to use with good psychometric properties as a screening or diagnostic tool for suicidal behavior in adolescents that can be used for general evaluation or psychopathological examination [54].

## 5. Conclusions

Suicidal ideation prevailed in almost two-thirds of Iraqi students. Univariable testing, multivariable analyses, and structural modeling yielded congruent results. Significant risk factors primarily included the PHQ-9 score, personal history of mental illness, previous psychiatric consultation, and family history of suicidality. These results agree with Durkheim’s theory on the social roots of suicide and highlight risk factors that psychiatrists and clinical psychologists could monitor among people with suicidal ideation and potential parasuicide victims.

Given the findings of our study and other studies conducted in Muslim countries, it can be admitted that determining the scale and predictors of suicidal ideation and behavior in communities practicing Islam is difficult for researchers. Therefore, we believe any research conducted for this purpose deserves recognition, mainly because it can contribute to a complete exploration of this topic and, simultaneously, the preparation of adequate preventive interventions.

## Figures and Tables

**Figure 1 ijerph-20-01795-f001:**
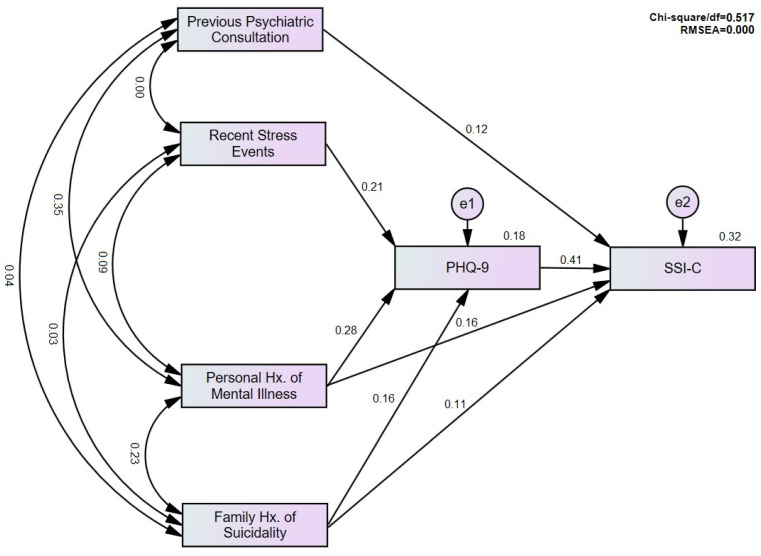
Path analysis and structural modeling. RMSEA: Root Mean Square Error of Approximation.

**Table 1 ijerph-20-01795-t001:** Chi-squared tests for PHQ-9 and SSI-C risk factors.

Risk Factor	Outcome	χ^2^	Sig.
Age	PHQ-9	1.12	0.539
	SSI-C	3.72	0.123
Gender	PHQ-9	1.38	0.240
	SSI-C	1.53	0.216
College Affiliation	PHQ-9	2.81	0.094
	SSI-C	<0.01	0.979
Marital Status	PHQ-9	0.74	0.389
	SSI-C	1.54	0.214
Religious Affiliation	PHQ-9	0.02	0.891
	SSI-C	1.94	0.164
Year of Study	PHQ-9	0.34	0.560
	SSI-C	1.00	0.318
Residence	PHQ-9	1.43	0.231
	SSI-C	0.70	0.406
Previous Psych. Consultation	PHQ-9	3.95	**0.047**
	SSI-C	0.70	0.403
Personal History of Mental Illness	PHQ-9	39.76	**<0.001**
	SSI-C	19.82	**<0.001**
Personal History of Chronic Illness	PHQ-9	1.68	0.195
	SSI-C	<0.01	0.956
History of Self-Medication	PHQ-9	0.56	0.455
	SSI-C	0.09	0.769
Recent Stress Events	PHQ-9	12.04	**0.001**
	SSI-C	1.71	0.190
Family History of Mental Illness	PHQ-9	2.78	0.095
	SSI-C	1.25	0.263
Family History of Suicidality	PHQ-9	23.29	**<0.001**
	SSI-C	7.51	**0.006**
PHQ-9	PHQ-9	---	---
	SSI-C	48.34	**<0.001**

Significant *p*-values are in bold fonts.

## Data Availability

The authors abide by an open-access policy concerning medical research data. All data will be available upon justifiable request from the corresponding author for three years following the publication.

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
