# Peer review of "Suicidal Ideation in Iraqi Medical Students Based on Research Using PHQ-9 and SSI-C"

_ijerph, 2023, doi:10.3390/ijerph20031795_

Round 1

Reviewer 1 Report

It is an interesting issue that in particular medical students are prone to depression and have suicidal ideations, but why are they called “patients” in the abstract?

In the introduction Edwin Schneidman`s definition of suicide is important but without the original reference. His typology of suicide is mentioned and listed but without the original reference.

Material and Method: how was asked for self-medication - definition?  And how did the authors measure stress – by live events?

Results: The presentation of the results in particular some tables are (1,2,3) are overloaded and redundant – but the main data are missing. Table 1 is unclear.

The first results tables should contain the sample description – and the Chi² tests of independence between demographics and the two psychological measures, presented in table 4. Table 1, 2 and 3 could be reduced to the most important aspects. Figure 1 and figure 2 are redundant.

The multivariate presentation and figure 3 are ok, but the supervised machine learning chapter and the figure 4 and 5 don’t add important information.

The discussion contains long descriptions repeating contents of the introduction and belong there. The real discussion starts at page 19, line 512, and it is important that not only results from western students are taken into account.

The additional non published material is also redundant in particular the figures of the categorial presentation of the PHQ-9.

The comparison of depression – the scores or the categories - between dentistry, medicine and pharmacy could be interesting

Author Response

Dear Sir/Madam,

Thank you very much for dedicating your time and efforts to peer review and critically analyze our manuscript. I am attaching a response in compliance with your comments and queries.

Best regards,

Ahmed.

Reviewer 2 Report

The manuscript “Suicidal ideation in Iraqi medical students based on research 1 using PHQ-9 and SSI-C: describes a survey study of suicidal ideation and their risk factors in a moderately sized sample of  medical students in Iraq. Suicidal ideation is measured by the Beck’s Scale for Suicidal Ideation, while depression was measured by PHQ-9.

While the paper in general is clearly written, I find the terminology used to classify  students, namely lower-risk suicidal ideation and higher-risk suicidal ideation,  misleading. First of all, a 0 value on the Beck scale indicates no ideation, thus it is misleading to name it ideation at all, thus the name low ideation does not fit (no- or low-ideation would be a better choice). Second, using a cutoff of 2 on the scale for high ideation is not justified by any references and is not supported by existing literature.  Grouping together subjects with a score of 2 and 38  and calling them all high risk does not make sense unless the goal was to compare ideator to non-ideators.  In fact, the study  with the present cutoffs could have been more believably presented as one comparing ideator and non-ideator subjects – there could be an argument that values of 1 on the Beck could be called negligible.

 A major question is, given that PHQ-9 does have an item, item 9, on suicidality, why  the distribution of answers on that item was not described and analyzed , and its association with the Beck score discussed?

Overall, the paper suffers from a lack of focus - it consists of long description of survey results in the text that would be better summarized  in one or two tables. The tests performed are mostly not significant, even without adjustment for multiple testing, and so it is hard to see what new conclusions one would draw- the importance of family history for predicting suicidality is already known.

Tables and figures actually included are also minimally informative, as the presentation of demographic information in Table 1  on age  by sex emphasizes meaningless detail and a multitude of  types of summary statistics over practically significant information on factors that are known to be associated with suicide risk, such as rural/urban location, although regions are listed in the text. In Table 2, means or proportions by group are also not presented.

Author Response

(The authors gave the same response as above.)

Reviewer 3 Report

First of all, I am grateful to have been chosen to read the article “Suicidal ideation in Iraqi medical students based on research 1 using PHQ-9 and SSI-C” Suicidal behavior is a worldwide public health problem that demands research to improve its management at all stages of life, but especially in adolescence. The article presented addresses this issue in young students and relates suicidal behavior to other important variables as risk factors.

The following are some areas for improvement in the text

-          I consider it necessary to reduce the number of keywords appearing in the text.

-          “According to the world mental health survey initiative, the lifetime prevalence of suicidal ideation, planning, and attempts is 9.2%, 3.1%, and 2.7%, respectively, while 60% of transitions from ideation to actual planning take place within the first year from the onset of ideation” more current data are available. I recommend reading:

o   Lim, K. S., Wong, C. H., McIntyre, R. S., Wang, J., Zhang, Z., Tran, B. X., ... & Ho, R. C. (2019). Global lifetime and 12-month prevalence of suicidal behavior, deliberate self-harm and non-suicidal self-injury in children and adolescents between 1989 and 2018: a meta-analysis. International Journal of Environmental Research and Public Health16(22), 4581. https://doi.org/10.3390/ijerph16224581

-          The introduction does not contain all the information that is later analyzed. More emphasis needs to be placed on the importance of early detection, on the related variables to be analyzed later, on the use of validated questionnaires, etc.

-          Materials and Methods

o   Why do you use Beck`s cut-off point of and not the more recent use of Beurs and coworkers?

o   It would be useful to justify the use of these instruments and their reliability in recent studies with adolescents.

o   It could be interesting to use more recent scales to assess suicidal behavior in all its facets. I recommend reading the following articles for this purpose:

§  Díez-Gómez, A., Sebastián-Enesco, C., Pérez-Albéniz, A., & Fonseca Pedrero, E. (2021). Suicidal behavior assessment in adolescents: Validation of the SENTIA-Brief scale. Actas Españolas de Psiquiatría, 49, 24-34.

§  Fonseca-Pedrero, E., & Pérez de Albéniz, A. (2020). Evaluación de la conducta suicida en adolescentes: A propósito de la escala Paykel de suicidio. Papeles del Psicólogo41(2), 106-115.

o   Table 1 does not describe the variables that have been collected, such as age, personal history, years of study, etc.

o   Table 1 it is not necessary to show so many values for age, the interquartile range, maximum, etc., is irrelevant information for the study.

-          Results:

o   It is not necessary to indicate “p-value”.

o   Information that is already contained in Materials and Methods is repeated.

o   Many variables that have not been justified in the introduction, such as the relationship with marital status, religion, etc., are included.

-          To thoroughly revise the reference section and the reference in the text to conform to the journal's standards.

-          Revise the format of tables and figures to conform to the journal format.

-          Many theories of the approach to suicidal behavior are discussed, but the relationship with the proposed study is not seen, nor are the most current models discussed in more depth.

-          In the discussion section, much information is introduced that should appear in the introduction and this section should focus on relating the results obtained with other studies.

Author Response

(The authors gave the same response as above.)

Round 2

Reviewer 1 Report

In the abstract the students are still called “patients”

The content is ok, but the presentation of the results is still confusing. P4-p6: these aspects should be included in  meaningful tables e.g. S1 and S2 in the supplementary (1) not the figures! – the arguments that the other reviewers wanted not so many tables are related to the meaningless tables and figures of version 1.

Very positive is the literature including papers about cultural risk factors for suicide ideation and suicide attempts among Muslim students

Reference 4 contains Canadian journal of psychiatry twice

Author Response

Peer Reviewer #1

  • In the abstract, the students are still called "patients".

Response: Dear Sir/Madam, we apologize for this mistake. We replaced the word "patients" with "students" within the abstract.

  • The content is ok, but the presentation of the results is still confusing. P4-p6: these aspects should be included in meaningful tables, e.g., S1 and S2 in the supplementary file, not the figures! – the arguments that the other reviewers wanted not so many tables are related to the meaningless tables and figures of version 1.

Response: Dear Sir/Madam, thank you for your feedback. We proofread and edited the results per your comments. We also edited the pages you highlighted and included details concerning those descriptive parameters within the supplementary file, including tables S1, S2, and Figure S1. Further, we provided details concerning students' affiliation to different colleges per your remark from round #1 of the peer review: "The comparison of depression – the scores or the categories - between dentistry, medicine and pharmacy could be interesting". We genuinely believe your intelligent critical appraisal made our research more coherent and the revised version more impactful. Once again, thank you for dedicating time and effort to peer-reviewing our article.

  • Very positive is the literature, including papers about cultural risk factors for suicide ideation and suicide attempts among Muslim students

Response: Dear Sir/Madam, thank you for your kind words. Our coauthor from Poland, Prof. Marek Motyka, addressed this section and information concerning the cultural risk factors for suicide ideation and suicide attempts among Muslim students.

  • Reference 4 contains the Canadian journal of psychiatry twice

Response: Dear Sir/Madam, thank you for your keen observation. We edited the bibliographic citation (reference #4) as per your comment.

Reviewer 2 Report

The authors have made many changes following recommendations,  as a result the manuscript is much improved.  The addition of the results on the 9th item of PHQ9 increases interest as it provides additional nuance as to frequency of suicidal ideation, also the analysis results in more  significant findings. The authors also convincingly argued in their response and new addition to the manuscript for the originality and significance of the findings.  I have no further objections.

Author Response

Dear Professor Dr. Tchounwou,

Dear Peer Reviewer,

Sir, we would like to submit the second revised version of the manuscript "Suicidal ideation in Iraqi medical students based on research using PHQ-9 and SSI-C." to your esteemed journal. We complied with each element of the peer review reports of the respected reviewers. Kindly find our detailed response below. We introduced minor revisions to the full-text article using the track changes option in Office Word.

Peer Reviewer #2

  • The authors have made many changes following recommendations, and as a result, the manuscript is much improved. The addition of the results on the 9th item of PHQ9 increases interest as it provides additional nuance as to the frequency of suicidal ideation, also the analysis results in more significant findings. The authors also convincingly argued in their response and new addition to the manuscript for the originality and significance of the findings. I have no further objections.

Response: Dear Sir/Madam, thank you for your sincere efforts in peer-reviewing our article. We genuinely believe these revisions will enhance the scholarly quality of the paper, and promote higher readability and citations within the academic and scientific community. The information we added concerning the correlation between the 9th item of PHQ9 and Beck's scale potentiated the significance of the results. Once again, we appreciate your efforts and dedication to enhancing our research and its significance.

Finally, please do not hesitate to contact us should you require further revisions, and we would be pleased to address it appropriately.

Best regards,

Dr. Ahmed Al-Imam, the corresponding author.

Reviewer 3 Report

The corrections indicated have been introduced in the text, therefore I consider it to be a work suitable for publication.

Author Response

Dear Professor Dr. Tchounwou,

Dear Peer Reviewer,

Sir, we would like to submit the second revised version of the manuscript "Suicidal ideation in Iraqi medical students based on research using PHQ-9 and SSI-C." to your esteemed journal. We complied with each element of the peer review reports of the respected reviewers. Kindly find our detailed response below. We introduced minor revisions to the full-text article using the track changes option in Office Word.

Peer Reviewer #3

  • The corrections indicated have been introduced in the text. Therefore I consider it to be a work suitable for publication.

Response: Dear Sir/Madam, revising the article to comply with your critical analysis concerning our research has been a pleasure. Once again, thank you for your efforts.

Finally, please do not hesitate to contact us should you require further revisions, and we would be pleased to address it appropriately.

Best regards,

Dr. Ahmed Al-Imam, the corresponding author.
